# Phenolic Composition and Antioxidant Activity of *Tanacetum parthenium* Cultivated in Different Regions of Ukraine: Insights into the Flavonoids and Hydroxycinnamic Acids Profile

**DOI:** 10.3390/plants12162940

**Published:** 2023-08-14

**Authors:** Karyna Hordiei, Tetiana Gontova, Sonata Trumbeckaite, Maksym Yaremenko, Lina Raudone

**Affiliations:** 1Department of Botany, National University of Pharmacy, Valentinivska St. 4, 61002 Kharkiv, Ukraine; 95karisha95@gmail.com (K.H.); tetianaviola@ukr.net (T.G.); m.yaremenko@ukma.edu.ua (M.Y.); 2Department of Pharmacognosy, Lithuanian University of Health Sciences, Sukileliu Av. 13, LT-50162 Kaunas, Lithuania; sonata.trumbeckaite@lsmu.lt; 3Laboratory of Biopharmaceutical Research, Institute of Pharmaceutical Technologies, Lithuanian University of Health Sciences, Sukileliu Av. 13, LT-50162 Kaunas, Lithuania

**Keywords:** feverfew, *Tanacetum parthenium*, antioxidant activity, phenolic compounds, HPLC–PDA

## Abstract

*Tanacetum parthenium*, also known as feverfew, is rich in bioactive compounds, namely sesquiterpene lactones, flavonoids, and volatile oils. Sesquiterpene lactones possess anti-migraine activity, while phenolic compounds possess anti-inflammatory and antioxidant action. Phytochemical composition determines the pharmacological activity and so profiling is essential in quality assessment. The study aimed to evaluate cultivated feverfew plants’ phenolic profiles and antioxidant activity. Eleven phenolic compounds were identified in the samples of feverfew in Ukraine. Hydroxycinnamic acids predominate in the quantitative content of all the samples, namely chlorogenic acid, 3,5-dicaffeoylquinic acid, 3,4-dicaffeoylquinic acid and 4,5-dicaffeoylquinic acid. The total content of flavonoids ranged from 0.8 to 2.6%; the content of hydroxycinnamic acids varied from 3.3 to 6.5%. The obtained data testify to the prospects of using Ukrainian feverfew as a raw material with a significant content of phenolic substances to develop new herbal medicines.

## 1. Introduction

Feverfew (*Tanacetum parthenium* (L.) Schultz Bip.)—is a representative of the Asteraceae family and one of the most prospective and insufficiently studied species of the *Tanacetum* L. genus, the herb of which is included as a raw material in the European Pharmacopoeia, United States Pharmacopeia, American herbal Pharmacopoeia, British Pharmacopoeia and the state Pharmacopoeia of Ukraine [1,2,3,4,5]. This perennial herb is native to the Balkan Peninsula and is widely cultivated in European countries and in Ukraine. Externally, the feverfew herb is recommended for treating psoriasis and dermatitis accompanied by itching, as well as for dressing wounds and mouthwash after dental surgery [6]. Feverfew raw materials have anti-inflammatory, cardiotonic, antipyretic and antispasmodic effects [7,8]. Over the past 10 years, biologically active preparations containing feverfew raw materials have been developed for the treatment and prevention of migraines [9]. Studies also confirm the anticancer activity of feverfew extracts [10,11,12].

However, feverfew has gained significant interest from scientists worldwide due to its antimigraine activity, which is primarily attributed to sesquiterpene lactones [13]. Migraine is known to be a significant reason for treatment costs and a cause of disability.

The primary objective of scientific uncertainty is to identify safe and effective medications to prevent migraine attacks. Over the past 10 years, various preparations based on this raw material have been developed in Western countries [9]. In the pharmaceutical market of Ukraine, migraine drugs are represented only by triptans, which are not equally effective for all patients. *T. parthenium* holds excellent potential for prevention strategies. The environmental conditions of the cultivation area, including climatic and edaphic factors, significantly impact the complex of bioactive compounds present in raw materials. Considering that no recent studies of the raw materials of the Ukrainian feverfew have been conducted, it was decided to carry out a comprehensive pharmacognostic analysis of this plant. This analysis will enable us to elucidate the natural potential of the feverfew plant.

The chemical profiles of the raw materials of feverfew are mainly characterized by the sesquiterpene lactones (parthenolide, artecanin, germacranolide, etc.) Another essential group is formed by the phenolic compounds. These are mainly composed of hydroxycinnamic acids (chlorogenic acid, dicaffeoylquinic acids, chicoric acid, etc.) and flavonoids (6-hydroxykaempferol-3,6-dimethyl ether, 6-hydroxykaempferol-6-trimethylether, 6-hydroxykaempferol-3,7,4’-trimethyl ether (tanetin), quercetagetin-3,6-dimethyl ether, quercetagetin-3,6,3′-trimethyl ether, quercetin, apigenin, apigenin-7-glucuronide, luteolin, luteolin-7-glucuronide, and chrysoeriol, santin) [14,15,16,17,18].

The pharmacognostic study of the raw materials of the Ukrainian feverfew proceeded with the quantification of the primary marker of raw material—parthenolide—in the seven samples from different regions of Ukraine. Based on the results, the parthenolide content in the studied samples varied from 0.16% to 0.39%, exceeding that of the UK’s raw material (classified as Cfb according to the Köppen climate classification). In the UK, the mixture of leaves and flowering tops contained 0.18% parthenolide, while the content in leaves and flowering tops was 0.09% and 0.27%, respectively [19]. Regarding feverfew raw material collected from different regions of Mexico (Cfb), the parthenolide content ranged from 0.25% to 0.28% [20]. Indeed, the geographical phytoprofiles demonstrate the way in which chemical polymorphism depends on the climatic, edaphic and genotypic characteristics. Plant extracts contain multiple chemical origin compounds, and it is crucial to determine the compounds that contribute to the therapeutical effects of the plant extracts. This paper focuses on the plant materials’ phenolic composition, which exhibits considerable variation according to the literature [16,17,18,21]. The content of flavonoids in the powder from the aerial parts of *T. parthenium* from Israel (Csa) was 0.54% [22]. *T. parthenium* leaves collected in India (Aw) contained 1.9% of flavonoids and, in the samples collected in France (Cfb), contained 1.1% [23,24]. The content of flavonoids in raw material from Iran (Bwh) was only 0.11 % with the following identified compounds: kaempferol naringenin apigenin, luteolin, quercetagetin, tanetin, santin, chrysoeriol, jaceidin and centaureidin [21]. The total content of phenolic compounds in raw materials collected in Romania (Cfb) amounted to 3.48 g/100 g, of which 1.27 g/100 g accounted for flavonoids and 1.30 g/100 g for caffeic acid derivatives. The common markers in the profiles are apigenin, luteolin, quercetin, chlorogenic, caffeic and gentisic acids [25]. Thus, in areas with a temperate climate, without a dry season and with a warm summer (Cfb) and in an area with a tropical climate, the savannah (Aw) plants accumulated more phenolic substances than in desert areas with a hot climate (Bwh).

Considering the variability in phenolic composition and antioxidant activity found in feverfew raw materials, as indicated by the literature, it becomes crucial to investigate the phenolic variation in feverfew raw materials from Ukrainian habitats. Moreover, it is essential to identify potential markers that can be used for standardization and the development of herbal medicines derived from the feverfew herb. Utilizing cultivated raw materials provides several benefits when compared with wild plants. These advantages include ensuring accurate identification, achieving greater uniformity in raw materials, and obtaining higher bioactive compound content influenced by sun exposure, watering, harvesting, and drying time. Moreover, cultivating raw materials allows for multiple harvests throughout the year and enables the use of enclosed areas for standardized cultivation, ensuring consistency in the chemical composition. Ukraine’s territory holds promise for the cultivation of medicinal plants due to several key advantages. Notably, it benefits from a temperate–continental climate (Dfb), offering around 230 clear sunny days annually. Moreover, Ukraine boasts expansive chernozem soils, known for their exceptional fertility. These factors, among others, contribute to the favourable conditions for cultivating medicinal plants in the region [26].

This study aimed to comprehensively investigate the phenolic compounds and antioxidant activity present in feverfew herbs collected from various regions of Ukraine. The purpose of this research was to provide a foundation for the future development of medicines based on feverfew.

## 2. Results and Discussion

Determining the total amounts of phenolic compounds derived from phenolic sources is crucial due to their significant role as antioxidants. These compounds play a vital role in scavenging free radicals and reactive oxygen species, thereby maintaining redox balance. Oxidative stress is closely associated with various inflammatory processes, including neuroinflammation in migraines is implicated in developing such conditions [9,27]. The etiopathogenesis of migraines is influenced by inflammation and the balance between oxidants and antioxidants. It has been proposed that migraine attacks act as a neuroprotective response to oxidative stress, and thus, phenolic antioxidants could potentially reduce the severity of migraines [28,29].

Studies confirm the preventative and protective effects of phenolic compounds and particularly caffeoylquinic acids, on neurodegenerative diseases [30,31]. Giacomo et al. determined that the anti-inflammatory and neuromodulatory effects of *T. parthenium* and *Salix alba* extracts are linked to their ability to counteract harmful free radicals. Additionally, the use of *T. parthenium* water extract resulted in significant changes in the rat cortex by reducing the levels of neurofilament and myelin-associated glycoprotein [32]. The concerns about potential neurotoxic effects emphasize the need to understand the phenolic profile and precise mechanisms of individual phytochemical markers. This is important due to the possibility of antioxidant compounds having pro-oxidant effects.

Scientific evidence has revealed that groups of key phenolic compounds in *Tanacetum* species are flavonoids and hydroxycinnamic acids [7]. Their evaluation provides valuable information on the total amounts of these compounds, allowing for a comprehensive understanding of their presence and activity in the *T. parthenium* materials. Table 1 presents our results of the quantitative analysis of flavonoids, hydroxycinnamic acids, and radical scavenging activity in selected samples of the feverfew herb.

The flavonoid content in the examined samples of the feverfew herb ranged from 0.79% to 2.65%. The greatest content of flavonoids was determined in the sample from the Kyiv region (870)—2.65 ± 0.03%. The lowest content was determined in the studied samples from the Cherkasy (865) and the Zhytomyr regions (864)—0.87 ± 0.02% and 0.79 ± 0.04%, respectively. There was no significant difference between samples 867 and 869. The determined amounts of hydroxycinnamic acids in the research samples ranged from 3.34% to 6.47%. The highest content of hydroxycinnamic acids was detected in the sample from the Poltava region (869)—6.47 ± 0.03%. The lowest amounts of hydroxycinnamic acids were in samples from the Cherkasy and the Zhytomyr regions—3.56 ± 0.02% and 3.34 ± 0.01%, respectively.

Radical scavenging activity ranged from 86.49 ± 1.07 µmol/g to 127.89 ± 1.04 µmol/g. The average value of radical scavenging in samples from different regions was 111.72 µmol/g. The obtained results clearly show that the highest activity was determined in samples 867 (126.67 ± 6.65 µmol/g) and 870 (127.89 ± 1.04 µmol/g). There were no significant differences between samples 866 and 869. The lowest radical scavenging activity was detected in sample 865 (86.49 ± 6.91 µmol/g). A significant correlation was determined between the total amounts of flavonoids and radical scavenging activity—*R*^2^ = 0.57; *R* = 0.75; *p* < 0.05. Total hydroxycinnamic acids and radical scavenging activity showed a correlational trend (*R*^2^ = 0.39; *R* = 0.62), although insignificant. Results suggest that flavonoids could be the predominant contributors to the antioxidant activity of the raw materials of *T. parthenium*.

The radical scavenging activity study results demonstrate that the feverfew herb samples with the highest flavonoids and hydroxycinnamic acids (866, 867, 869, and 870) exhibited the most pronounced antioxidant activity. In this case, sample 864 exhibits significantly higher radical scavenging activity than sample 865, despite having fewer tested compound groups. Sample 865 has a similar quantitative content, while sample 868 has a 1.5-fold higher content of both flavonoids and hydroxycinnamic acids. This difference between the content of phenolic compounds and antioxidant activity may be due to several factors. First, the total phenolic content does not contain all antioxidants. Secondly, the synergism between antioxidants in the mixture should be considered, which means that the antioxidant activity not only depends on the concentration but also on the structure and interaction between antioxidants. Some phenolic compounds may exhibit potent antioxidant activity even at low concentrations, while others may have limited or no antioxidant activity despite being present in high concentrations [33]. On the other hand, the antioxidant activity of a sample can be affected by other non-phenolic components present in the extract. These non-phenolic components may contribute significantly to the overall antioxidant capacity, leading to a disparity between total phenolic content and antioxidant activity [34]. Babich et al. determined the radical scavenging and reducing activities of *T. parthenium*, specifically focusing on chlorogenic acid and luteolin-7-glucoside as key markers in the phytochemical profile. However, they did not assess the total antioxidant activity of the prepared extracts. [35]. Wu et al. elucidated that the flavonoids luteolin and apigenin may possibly be the main contributors to total antioxidant activity [36]. Plant extracts’ antioxidant activity could not be defined solely based on their total phenolic content, so it is essential to characterize the full phenolic profile and identify its peculiarities.

According to our results, three flavonols (quercetin, kaempferol-3-rutinoside and o-methylated flavonol santin), one flavone (apigenin), and six hydroxycinnamic acids (chlorogenic acid, 4-o-caffeoylquinic acid, 3,4-dicaffeoylquinic acid, 3,5-dicaffeoylquinic acid, 4,5-dicaffeoylquinic acid and neochlorogenic acid) were identified in all of the tested samples. Flavonoid santin, a specific marker for the feverfew, was detected and quantified in all of the samples except for the raw material collected in the Kyiv region. Ellagic acid was also identified in five series (864, 865, 868, 869, 870). The results of the study are presented in Table 2 and Figure 1.

The HPLC analysis revealed significant variations in the content of identified components within the feverfew herb, depending on the cultivation location. The total content of the identified compounds ranged from 12,184.79 µg/g to 23,701.62 µg/g DW. The results highlight the observation that the tested samples of feverfew exhibit a considerably higher total content of hydroxycinnamic acids compared with flavonoids. Among the analyzed series, the smallest disparity in content between these two compound groups was determined in series 869, where the hydroxycinnamic acids were seven times more abundant than flavonoids. Conversely, the largest discrepancy was observed in sample 870, with the flavonoid content being 18 times lower than hydroxycinnamic acids. These findings underscore the significant differences in the composition of feverfew samples, suggesting the considerable impact of cultivation location on the content of identified compounds.

In the profile of hydroxycinnamic acids, 3,5-dicaffeoylquinic acid, chlorogenic acid and 3,4-dicaffeoylquinic acid were the dominant components in all samples. Thus, 3,5-dicaffeoylquinic acid accounts for 43% to 50% of all quantified substances, which is quite significant compared with the results of a study of raw materials collected in China (Dwb), where the content was 31.78%. The content of 3,5-dicaffeoylquinic acid ranged from 56.61 ± 0.30 µg/mL to 120.68 ± 0.26 µg/mL, the highest value was observed in sample 870 and exceeded a 2.7-fold increase over the results of studies of raw materials from Brazil (Cfa), which indicated 44.72 ± 0.30 μg/mL [37].

Chlorogenic acid as a secondary metabolite is formed by the esterification of one or more trans-cinnamic acid derivatives with quinic acid. This compound has numerous pharmacological effects, with antioxidant, anti-inflammatory, antineoplastic and antiviral activity as the most expressed [38,39,40]. The content of chlorogenic acid was significantly higher according to the results of our study, namely from 28.07 ± 0.16 µg/mL (864) to 73.56 ± 0.82 µg/mL (866), in contrast with the raw materials collected in Brazil (Cfa), where its content was 7.91 ± 0.18 µg/mL [38]. Based on the data of the analyzed raw materials collected in China (Dwb), the content of chlorogenic acid was 12% of all identified compounds. However, our results show a significantly higher percentage, ranging from 18% to 28%, depending on the place of harvest [41].

The content of 3,4-dicaffeoylquinic acid to the total content of quantified substances was 10–17%, which is slightly less than in the raw materials from China (Dwb), which was 23.3% [41]. However, the content of 3,4-dicaffeoylquinic acid ranged from 19.821 ± 0.27 µg/mL (864) to 41.263 ± 0.285 µg/mL (870), which was 10–22-fold higher than in the raw materials collected in Brazil (Cfa), which was 1.85 ± 0.42 µg/mL [38]. The identified dicaffeoylquinic acids (3,4-dicaffeoylquinic acid and 3,5-dicaffeoylquinic acid) of the feverfew show pronounced activity against herpes simplex virus (anti-HSV-1 activity), an observation that has been proven by numerous in vitro and in vivo studies [38,42,43].

The content of 4,5-dicaffeoylquinic acid was 1.25 times higher in the raw materials cultivated in Brazil (Cfa), which was 17.7 ± 1.21 µg/mL, compared with our result of 14.12 ± 0.08 µg/mL [38]. Results suggest the caffeoylquinic acids as fingerprints of the *T. parthenium* profile, and the variation of their quantities highly depends on the growing climatic region.

Kaempferol-3-rutinoside and santin predominated in the quantitative profile of flavonoids. Thus, the content of kaempferol-3-rutinoside in the test samples varied from 4% to 9% and that of santin from 2% to 4% in the total content of the identified substances. Kaempferol-3-rutinoside has not been highlighted in the feverfew herb in the previous studies, although the flavonoid santin has already been found in raw materials from Israel (Csa) and China (Dwb). In the latter case, its content was 1.1% of the total amount of identified substances, which was a 2–4-fold decrease when compared with the Ukrainian raw materials [22,41]. The flavonoid santin is interesting as a pharmacologically active agent specific to the genus *Tanacetum*. It has also been detected in the herb *T. micropyllum*, collected in Madrid, Spain (Bsk). This compound has been shown to possess anti-inflammatory activity by inhibiting the macrophages’ functions in the inflammatory process. This compound significantly reduced the content of lipopolysaccharide (LPS), nitric oxide (NO) and prostaglandin (PGE). The selectivity of santin is worth noting in contrast with other flavonoids relative to COX-2 in LPS-stimulated macrophages, which gives significant advantages in treating inflammatory conditions [44].

The highest content of identified phenolic compounds was determined in sample 866, from the Sumy region (Dfb) (23,701.62 µg/g DW). This sample contained the greatest amounts of chlorogenic acid (6993.44 µg/g DW). The total content of identified compounds was also significant in samples 869 and 870 (23,655.93 µg/g and 22,464.56 µg/g DW, respectively). The high content of 3,5-dicaffeoylquinic acid (11,276.17 µg/g DW) and 4,5-dicaffeoylquinic acid (1324.93 µg/g DW) in sample 870 from the Kyiv region (Dfb) were determined, but santin was not identified. These findings highlight the influence of climatic conditions on the phenolic content of the samples, as different regions exhibited variations in the specific compounds present and their concentrations.

The highest amount of kaempferol-3-rutinoside and santin was identified in samples 866 and 869 (1663.24, 535.89 µg/g DW and 2065.29, 581.96 µg/g DW, respectively), which correlates with the obtained results of the spectrophotometric study of the total flavonoid content. The content of ellagic acid was the highest in sample 869 (165.60 µg/g DW). This compound was not identified in samples collected in Dnipro and Sumy regions.

The content of the flavone apigenin was quite high in the studied samples and ranged from 73.31 to 306.96 µg/g DW (from 0.3% to 1.3% of the total amount of identified substances). The apigenin content in the studied raw materials was significantly higher than in previous studies of similar materials harvested in Romania (Cfb), where the apigenin content was 9.71 µg/g DW. However, it was lower than the results obtained from raw materials sourced from China (Dwb), where the content was 5.56% [25,41].

It should be noted that the total content of all the identified and calculated compounds is the lowest in the samples collected in the Zhytomyr (Dfb) and Cherkasy (Dfb) regions (12,184.79 and 13,785.89 µg/g DW, respectively), which corresponds to the data obtained by spectrophotometric determination of the sum of hydroxycinnamic acids and flavonoids.

The cluster analysis divided the substances identified in the test samples into two statistically significant clusters (Figure 2). The first cluster included samples of the feverfew herb (864, 865 and 868) that contained the lowest amount of phenolic compounds, especially 3,5-dicaffeoylquinic acid, which accumulated about 30% less than in samples of the feverfew herb of the second cluster. The second cluster was characterized by the high sum average content of phenolic compounds, which ranged from 21 mg/g to 23 mg/g. Both clusters included three dominant substances: 3,4-dicaffeoylquinic, chlorogenic and 3,5-dicaffeoylquinic acids, accounting for up to 85% of the content of all compounds. These substances predominated in the profiles and could be applied in the standardization of the raw materials of the feverfew cultivated in Ukraine.

We also studied the influence of several climatic factors on the accumulation of phenolic substances. In the central regions of Ukraine (866, 867, 869 and 870), a higher content of phenolic substances was observed when the average air temperatures were higher. Additionally, the average annual temperature ranged in these areas from 9.5 °C to 10.1 °C, the annual average maximum temperature was 14.1–15.3 °C, the annual average minimum temperature was 4.5–5.1 °C, and the annual average humidity was 69.8–71.4%. In the samples from areas with a lower content of compounds (864, 865 and 868) the average temperatures were lower: average annual temperature was 9.1 °C to 9.9 °C, annual average maximum temperature was 13.5–14.3 °C, and annual average minimum temperature was 3.9–5.0 °C; however, a higher humidity was observed of 72.4–73.8%. Thus, in areas with higher air temperatures and lower humidity, phenolic compounds accumulate in greater quantities, an observation which also is in agreement with the published scientific data [45].

Overall, 3,5-dicaffeoylquinic acid, chlorogenic acid, 3,4-dicaffeoylquinic acid, kaempferol-3-rutinoside and santin could be markers for the phenolic profile of the feverfew herb collected from Ukraine.

## 3. Materials and Methods

### 3.1. Plant Materials

The samples of the feverfew herb were collected from 7 regions of central Ukraine in 2017 (Figure 3). The raw material was collected during the mass flowering of the plant (July–August). Conditions for the cultivation of the feverfew differed slightly in different areas: the soils of the experimental plots of 2, 4, 5 and 6 regions—low humus, deep chernozems, mechanical composition—light; for 1 and 7 regions—more humus, sod-podzolic, by mechanical composition—medium; for region 3—low humus, chernozems and gray podzolic forest, by mechanical composition—light [46]. The climate in all selected areas for cultivation is humid continental with warm summers (Dfb). The total annual solar radiation under average cloud conditions in region 1 is up to 3800 MJ/m^2^, and in all other regions is up to 4200 MJ/m^2^. The average annual rainfall in Ukraine decreases from 650–550 mm in the northern regions (1, 3, 7) to 540–500 mm in the central and eastern regions (2, 4, 5, 6) [47]. The size of the experimental plots was 15 m^2^, and the seeds were sown to a depth of 0.5–1 cm with a row spacing of 30 cm. Drying of the samples was performed by the air-shadow method according to the requirements of the pharmaceutical regulatory documents [48]. Samples for analysis were prepared by collecting and combining 3 individual plants as one sample from a specific area.

The samples of raw material are registered in the Ukrainian Scientific Pharmacopoeial Center for Quality of Medicines. They will be used to develop a national section for the monograph ‘*Tanacetum parthenium*’. Tetiana Gontova carried out the botanical identification of plant species. The herbarium vouchers were deposited at the National Herbarium of Ukraine (voucher specimen numbers: 864, 865, 866, 867, 868, 869, 870). To better understand the accumulation of phenolic compounds in the feverfew, we also analyzed the results obtained in different countries using Köppen–Geiger climate zones [49]. Further, the climatic zones are presented in parentheses.

### 3.2. Chemicals and Solvents

Water was purified using a Milli–Q system (Millipore, Bedford, MA, USA). Ethanol (96%) was obtained from Vilniaus degtine (Vilnius, Lithuania). Anhydrous acetic acid (99.8%), and hydrochloric acid (37%) were purchased from Sigma-Aldrich (Buchs, Switzerland). The following reagents were used: 2,2′-azino-bis(3-ethylbenzothiazoline-6-sulfonic acid) and diammonium salt (ABTS). The following standards were used: 6-hydroxy-2,5,7,8-tetramethylchroman-2-carboxylic acid (Trolox), quercetin, chlorogenic acid, apigenin, 4-o-caffeoylquinic acid, 3,4-dicaffeoylquinic acid, 3,5-dicaffeoylquinic acid, 4,5-dicaffeoylquinic acid, neochlorogenic acid, kaempferol-3-rutinoside, and ellagic acid from Sigma-Aldrich (Buchs, Switzerland) and santin from PlantMetaChem (Giesen, Germany). All the reagents and standards were of analytical grade. The stock solutions of phenolic compounds were prepared in 96% ethanol.

### 3.3. Total Flavonoids and Total Hydroxycinnamic Acids

Quantitative determinations of the total amount of flavonoids and the total amount of hydroxycinnamic acids in samples of the feverfew herb collected in 7 regions of Ukraine were carried out by spectrophotometry using unified methods described in the State Pharmacopoeia of Ukraine.

Quantitative determination of the sum of hydroxycinnamic acids in the samples was performed by a unified pharmacopoeial spectrophotometric method based on the determination of hydroxycinnamic acids after reaction with sodium nitrite and sodium molybdate under the requirements of NPhU 2.0 monograph 2.2.25. Quantitative determination of the content of the amount of flavonoids in the samples was performed by a unified pharmacopoeial spectrophotometric method based on the determination of flavonoids after reaction with a mixture of boric acid P and oxalic acid P in a mixture of formic acid P and acetic acid P [50].

### 3.4. ABTS Radical Cation Decolorization Assay

The radical scavenging activity was evaluated by the scavenging of ABTS radical cation. A standard calibration curve obtained using a standard sample of trolox was constructed. ABTS radical cation scavenging activity of extracts was obtained from the regression equation: y = 0.0002x + 0.0181 (*R*^2^ = 0.9976) and expressed as antioxidant trolox equivalents (TE) per gram of material [51].

### 3.5. Qualitative and Quantitative Analysis by HPLC–PDA Method

HPLC analysis was performed using a “Waters e2695 Alliance system” (Waters, Milford, MA, USA) with a photodiode array detector “Waters 2998” according to the HPLC–PDA method for phenolic compounds. Briefly, the “ACE” (ACT, UK) column (C18,150 mm × 4.6 mm, particle size 3 μm) was used. The gradient consisted of eluent A (0.05% trifluoracetic acid) and B (acetonitrile) and followed: 0–5 min—12% B, 5–50 min—12–30% B, 50–51 min—30–90% B, 51–56 min—90% B, and 57 min—12% B with the flow rate of 0.5 mL/min and injection volume of 10 μL. The analyte and reference compound retention time and UV absorption spectra were used for peak identification. The calibration curves of reference compounds were constructed. Contents of phenolic acids were estimated at a wavelength of 325 nm, while the contents of flavonoids were estimated at a wavelength of 350 nm.

### 3.6. Statistical Analysis

Statistical analysis was performed using SPSS 26.0 (SPSS Inc., Chicago, IL, USA) and Microsoft Office Excel 365 (Microsoft, Redmond, WA, USA). All measurements were made in triplicate, and results expressed as mean ± standard deviation (SD). Linear regression analysis was performed to calculate the concentration-response relationship of each investigated compound by ABTS assay. Correlations were tested by using the Pearson correlation test. One-way analysis of variance was performed by ANOVA test. Significant differences between means were determined by Tukey HSD multiple comparison test. The *p*-values less than 0.05 were considered statistically significant.

## 4. Conclusions

This study showed the significant differences between the phenolic composition of the feverfew herb collected in Ukraine and the results of studies from other countries. Eleven compounds were identified, indicating a significant qualitative composition of phenolic compounds in the studied samples. Of the identified compounds, hydroxycinnamic acids predominate in quantitative content in all the samples, namely chlorogenic, 3,5-dicaffeoylquinic acid, 3,4-dicaffeoylquinic acid and 4,5-dicaffeoylquinic acid. Certain substances can be important tools for assessing the quality of the feverfew herb. The quantitative content of compounds of phenolic nature is quite significant. Recent scientific investigations have unveiled their potential role in mitigating migraine symptoms. These compounds exhibit diverse pharmacological activities, including anti-inflammatory, antioxidant, and neuroprotective effects. By targeting key pathways implicated in migraine pathogenesis, phenolics coupled with terpene-origin compounds hold promise for migraine prevention and management. Moreover, understanding the phytogeographical profiles of *T. parthenium* can contribute to natural resource management and cultivation strategies. The targeted homogeneous phytochemical composition can be ensured by identifying regions with favourable environmental conditions. The results of this study emphasize the importance of considering climatic zones when evaluating phenolic content in plants. Understanding these variations can affect agricultural practices, food production, and even the potential health benefits associated with phenolic compounds. Further research in this field may provide valuable insights into the mechanisms behind these variations and their potential applications in various industries. The obtained data testify to the prospects of using Ukrainian raw materials of the feverfew as a raw material with a significant content of phenolic substances to create new herbal ingredients for medicinal preparations.

## Figures and Tables

**Figure 1 plants-12-02940-f001:**
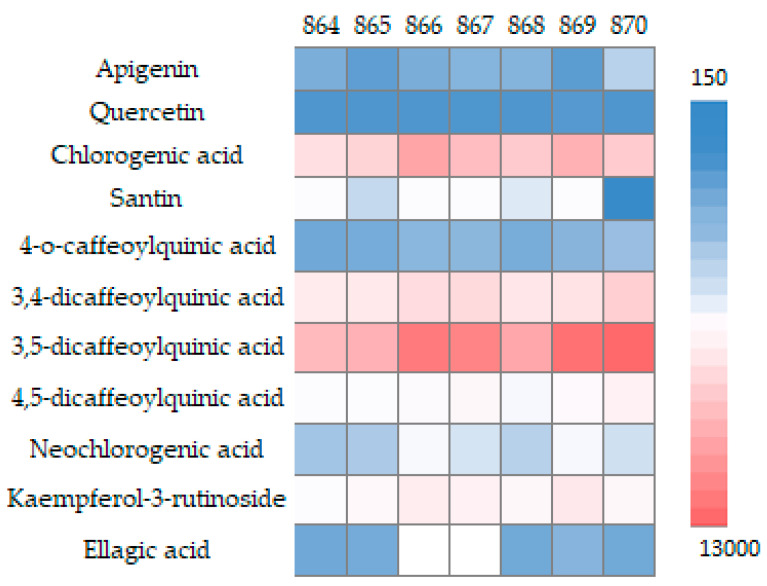
Heatmap visualization of the phenolic profiles (µg/g of DW) of *T. parthenium*.

**Figure 2 plants-12-02940-f002:**
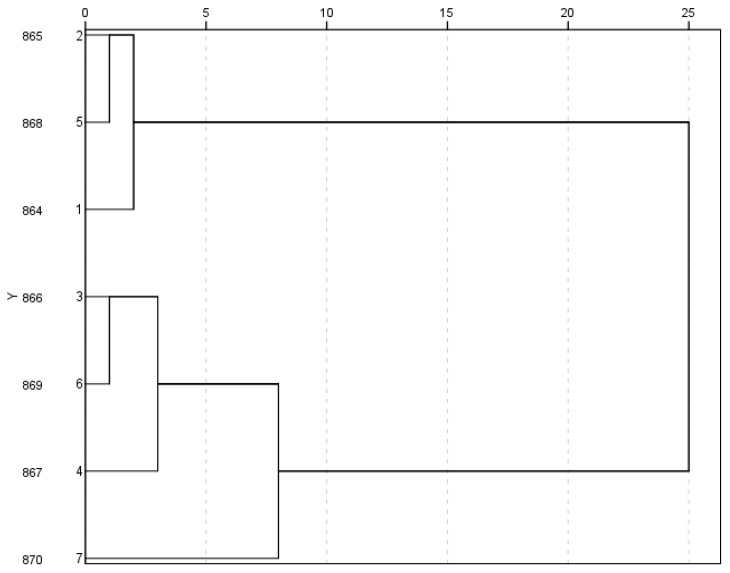
Dendrogram based on the amounts of phenolic compounds in samples of the feverfew herb.

**Figure 3 plants-12-02940-f003:**
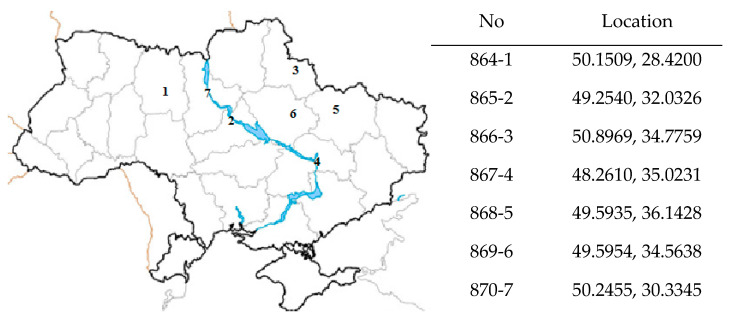
Registration numbers and cultivation areas of the samples of the feverfew herb.

**Table 1 plants-12-02940-t001:** Quantitative determination of the total amount of flavonoids (n = 3), hydroxycinnamic acids (n = 3) and radical scavenging activity (n = 5) in samples of the feverfew herb.

Research Sample	The Total Content of Flavonoids, %	The Total Content of Hydroxycinnamic Acids, %	The Radical Scavenging Activity, ABTS µmol/g
864	0.79 ± 0.04 b	3.34 ± 0.01 a	103.19 ± 1.07 b
865	0.87 ± 0.02 c	3.56 ± 0.02 b	86.49 ± 6.91 c
866	1.92 ± 0.03 d	5.10 ± 0.07 c	124.97 ± 3.21 a
867	1.84 ± 0.02 a	4.64 ± 0.03 d	126.67 ± 6.65 a
868	1.58 ± 0.03 e	4.54 ± 0.01 e	89.27 ± 1.74 c
869	1.87 ± 0.03 a	6.47 ± 0.03 f	123.53 ± 4.50 a
870	2.65 ± 0.03 f	4.46 ± 0.02 g	127.89 ± 1.04 a

Values are means ± SD. Contents marked without the same letters (a, b, c) in the columns indicate statistically significant differences among series (*p* < 0.05).

**Table 2 plants-12-02940-t002:** Quantitative composition (µg/g of DW) of identified phenolic compounds in the *Tanacetum parthenium* (L.) Schultz Bip. herb depending on the regions of cultivation.

Compounds	864	865	866	867	868	869	870
Apigenin	140.45 ± 1.90 a ^1^	76.71 ± 0.49 c	136.88 ± 1.68 a	162.33 ± 0.83 b	160.76 ± 0.67 b	69.16 ± 3.92 d	286.87 ± 1.83 e
Quercetin	48.23 ± 0.44 a	46.50 ± 0.32 b	47.90 ± 0.28 a	44.89 ± 0.15 c	46.87 ± 0.24 b	58.22 ± 0.26 e	44.24 ± 0.29 c
Chlorogenic acid	2681.04 ± 24.12 b	3511.85 ± 31.10 c	6993.44 ± 86.57 d	5186.79 ± 45.21 e	4225.90 ± 21.25 a	6149.99 ± 16.97 f	4144.12 ± 2.59 a
Santin	488.24 ± 4.03 a	317.56 ± 3.68 b	535.89 ± 6.66 c	468.15 ± 2.12 d	382.65 ±1.35 e	581.96 ± 2.43 f	traces
4-o-caffeoyl-quinic acid	115.91 ± 0.68 b	129.33 ± 0.86 c	174.24 ±2.06 a	176.49 ± 1.59 a	134.89 ± 2.94 d	168.70 ± 0.38 e	209.49 ± 3.28 f
3,4-dicaffeoyl-quinic acid	1907.42 ± 31.43 a	2001.53 ± 43.08 a	2888.31 ± 42.31 b	3089.87 ± 18.11 c	2225.71 ± 33.05 d	2436.97 ± 39.89 e	3873.51 ± 23.55 f
3,5-dicaffeoyl-quinic acid	5402.57 ± 45.70 a	6077.21 ± 52.49 b	10159.95 ± 109.96 c	9423.89 ± 61.76 d	6853.07± 41.35 e	10799.61 ± 26.15 f	11276.17±16.74 g
4,5-dicaffeoyl-quinic acid	559.63 ± 4.82 b	459.90 ± 3.36 a	650.53 ± 6.03 c	891.59 ±4.36 d	447.42 ± 5.47 a	715.16 ± 1.56 e	1324.93 ± 6.85 f
Neochlorogenic acid	234.15 ± 1.94 b	256.80 ± 1.77 c	451.24 ± 5.94 a	359.96 ± 2.74 d	286.20 ± 0.79 e	445.28 ± 0.72 a	345.74 ± 1.08 f
Kaempferol-3-rutinoside	493.32 ± 7.42 a	783.22 ± 9.16 b	1663.24 ± 23.79 c	1285.69 ± 9.52 d	884.42 ± 6.74 e	2065.29 ± 14.40 f	840.52 ± 8.55 g
Ellagic acid	113.82 ± 0.85 b	125.27 ± 1.28 c	traces	traces	117.31 ± 0.22 a	165.60 ± 0.62 d	118.96 ± 0.38 a
**Total of all quantitated compounds**	**12,184.79**	**13,785.89**	**23,701.62**	**21,089.64**	**15,765.20**	**23,655.93**	**22,464.56**

^1^ Contents marked without the same letters (a, b, c, d, e, f, g) in the rows indicate statistically significant differences among samples series (*p* < 0.05); values are means ± SD (n = 3).

## Data Availability

All data generated during this study are included in this article.

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
