# Peer review of "Phenolic Composition and Antioxidant Activity of Tanacetum parthenium Cultivated in Different Regions of Ukraine: Insights into the Flavonoids and Hydroxycinnamic Acids Profile"

_plants, 2023, doi:10.3390/plants12162940_

Round 1

Reviewer 1 Report

Title: Phenolic Composition and Antioxidant Activity of Tanacetum parthenium Cultivated in Different Regions of Ukraine: Insights into the Flavonoids and Hydroxycinnamic Acids Profile

Authors: Karyna Hordiei et al.

  1. Check the abbreviations, the text format of full text, especially figures. 
  2. There are some grammatical errors in the text.
  3. The Figures and Tables should be rearranged.
  4. The discussion should be improved. 

There are some grammatical errors in the text.

Author Response

Reviewer 1: 

Check the abbreviations, the text format of full text, especially figures.

There are some grammatical errors in the text.

The Figures and Tables should be rearranged.

The discussion should be improved.

Response: We would like to thank you for your kind review and for taking the time and effort necessary to improve our manuscript. We found your comments very helpful and revised the manuscript accordingly. The abbreviations were revised. The grammatical errors were corrected. The numbering of figures and tables was rearranged. The discussion part was expanded, and certain paragraphs were rewritten. All changes have been indicated in red in the manuscript

Sincerely,

Lina Raudone on behalf of all authors.

Reviewer 2 Report

1.     Please revise the format of the keywords carefully, “;” should be used in keywords.

2.     Please pay more attention to grammar problems in the manuscript, such as page 3, line 122, “The content of flavonoids in the studied samples of the feverfew herb varies from 0.79 % to 2.65 %.”; page 4, line 143, “the highest activity is observed in samples…”.

3.     Please double check the format in the manuscript, such as page 3, line 134 “867 (126.67 ± 6.65 μmol/g) and 870 (127.89 ± 1.04 μmol / g)”, “μmol / g” spaces should be deleted; page 4, line 137 “the total amounts of flavonoids and radical scavenging activity – R2=0.57; R=0.75; p<0.05”, “R and p” should be italic.

4.     Please unify the format of graphs and tables in the text, such as page 3, line 114 “The esti-mated total amounts of the above-mentioned groups of compounds are presented in Table 1”; page 4, line 169 “The results of the study are presented in table 2 and figure 1”; page 8, line 266 “The cluster analysis divided the substances identified in the test samples into 2 sta-tistically significant clusters (Fig. 2).”

5.     I would suggest that the authors could consider dividing the results into different sections to make the logic of the manuscript clearer.

6.     Please check and revised reference carefully. The journal names are not uniform, the Latin names of plants are not italics and some items are lack of page numbers. Page 8, line 434; Page 9, line 450; Page 9, line 468.

Moderate editing of English language required

Author Response

Reviewer 2: 

Response: We would like to thank you for your kind review and for taking the time and effort necessary to improve our manuscript. We found your comments very helpful and revised the manuscript accordingly.

  1. Please revise the format of the keywords carefully, “;” should be used in keywords.

Response: the keywords were corrected.

  1. Please pay more attention to grammar problems in the manuscript, such as page 3, line 122, “The content of flavonoids in the studied samples of the feverfew herb varies from 0.79 % to 2.65 %.”; page 4, line 143, “the highest activity is observed in samples…”.

Response: a full grammar check was performed. The sentences were rearranged. The mistakes were corrected.

  1. Please double check the format in the manuscript, such as page 3, line 134 “867 (126.67 ± 6.65 μmol/g) and 870 (127.89 ± 1.04 μmol / g)”, “μmol / g” spaces should be deleted; page 4, line 137 “the total amounts of flavonoids and radical scavenging activity – R2=0.57; R=0.75; p<0.05”, “R and p” should be italic.

Response: The formats were revised, and the mistakes were corrected. The “R and p” were changed to italics.

  1. Please unify the format of graphs and tables in the text, such as page 3, line 114 “The esti-mated total amounts of the above-mentioned groups of compounds are presented in Table 1”; page 4, line 169 “The results of the study are presented in table 2 and figure 1”; page 8, line 266 “The cluster analysis divided the substances identified in the test samples into 2 sta-tistically significant clusters (Fig. 2).”

Response: the format of graphs and tables was revised, the mistakes were corrected, and the terms were unified.

  1. I would suggest that the authors could consider dividing the results into different sections to make the logic of the manuscript clearer.

Response: The results and discussion section were revised; certain sentences were rearranged and supplemented.

  1. Please check and revised reference carefully. The journal names are not uniform, the Latin names of plants are not italics and some items are lack of page numbers. Page 8, line 434; Page 9, line 450; Page 9, line 468.

Response: The reference list was revised, and the mistakes were corrected. The Latin names were italicized.

Sincerely,

Lina Raudone on behalf of all authors.

Reviewer 3 Report

Dear Authors

In this current manuscript, Raudone and co-workers have studied the phenolic composition and antioxidant profile of seven cultivars of feverfew. They chose selected areas for collecting samples including humid continental with warm. Among the compounds identified and quantified in all feverfew including 3-flavonols, santi (marker) and hydroxycinnamic acids. Finally, those findings based on HPLC analysis indicated that the among of compounds in the feverfew vary according to the place of sampling.

I think this work is very simple, intensive and good done, therefore I think is suitable for publication.

Author Response

Reviewer 3: 

In this current manuscript, Raudone and co-workers have studied the phenolic composition and antioxidant profile of seven cultivars of feverfew. They chose selected areas for collecting samples including humid continental with warm. Among the compounds identified and quantified in all feverfew including 3-flavonols, santi (marker) and hydroxycinnamic acids. Finally, those findings based on HPLC analysis indicated that the among of compounds in the feverfew vary according to the place of sampling.

I think this work is very simple, intensive and good done, therefore I think is suitable for publication.

Response: We would like to thank you for your kind review and for taking the time and effort necessary to evaluate our manuscript.

Sincerely,

Lina Raudone on behalf of all authors.

Reviewer 4 Report

The present manuscript entitled "Phenolic Composition and Antioxidant Activity of Tanacetum parthenium Cultivated in Different Regions of Ukraine: Insights into the Flavonoids and Hydroxycinnamic Acids Profile", describes the chemical composition and antioxidant evaluation of samples of Tanacetum parthenium obtained from various regions of Ukraine.

The analysis of the chemical composition of a medicinal plant is relevant to relate the phytochemical profile with the therapeutic properties. Thus, the present study is relevant and meets the scope of the journal.

Reading the manuscript showed many errors and adjustments that should be made in the text. The quality of writing is not good and, at times, it becomes a text that is heavy and confusing to read. In this way, I do not consider it to have the necessary quality for the publication of Plants. A comprehensive and rigorous review of the manuscript is required.

Below are the tweaks and corrections.

- In line 21, replace “chlorogenic” with ‘chlorogenic acid’;

- On line 23, replace “3.3 to 6.5.” by “3.3 to 6.5%.”;

- On line 25, I suggest replacing "create" with "develop”;

- In lines 31-32, the term "European, American, American herbal" is confused. Do the authors intend to write “American and European Herbal Pharmacopoeias”? Please review.

 - On line 42, the authors wrote "which is caused primarily by sesquiterpene lactones [13]". The study described in reference 13 contains pharmacological experiments on the plant extract only. There is a computer simulation of a constituent of the plant, which is a binding energy calculation. It does not allow concluding that it has pharmacological action. Therefore, the authors cannot claim that sesquiterpene lactones have antimigraine activity, based on this study. Please correct your writing.

 - On lines 46-47 authors wrote "Over the past 10 years, biologically active additives based on this raw material have been developed in Western countries [9]". However, reference 9 is a literature review published in 2004. So it is wrong to write that it is about the last 10 years. Please update the reference or revise the sentence.

 - In line 56, the term chrysanthemum is not sesquiterpene. It is the name of a genus of plant. Please correct.

 - On line 57, replace " (chlorogenic, dicaffeoylquinic, chicoric" with " (chlorogenic acid, dicaffeoylquinic acid, chicoric acid".

 - On line 59, correct "6-trimethylfether" and proofread all compound names in this paragraph. For example, tanetin is not 6-trimethyl ether ether; camphor, camphene, p-cymene, bornyl acetate are not essential oils. They are constituents of essential oils. It is very important that authors be more careful in writing this information. There are many errors in the Introduction of the manuscript. Please review the paragraphs rigorously.

 - On line 68, correct “tops was 0.18%, 0.09 in leaves”.

 - Referring to lines 108-115 of the first paragraph of Results and Discussion, authors wrote "The estimated total amounts of the above-mentioned groups of compounds are presented in Table 1. " Then, it presents a table with experimental data.

- This paragraph comments on data from other studies and cites their references. So does table 1 contain these data or data from the current study? If it is current study data, then authors should improve the text in a more enlightening way.

 - The numbers in Table 2 were written with a comma instead of a point. Por exemplo, Apigenin 140,45 ± 1,90. Please confirm that this is correct.

 - In the Plant materials section, authors should write the plant specimen code and deposit location. Please also inform that it was the taxonomist who made the identification of the species.

Author Response

Reviewer 4: 

The present manuscript entitled "Phenolic Composition and Antioxidant Activity of Tanacetum parthenium Cultivated in Different Regions of Ukraine: Insights into the Flavonoids and Hydroxycinnamic Acids Profile", describes the chemical composition and antioxidant evaluation of samples of Tanacetum parthenium obtained from various regions of Ukraine.

The analysis of the chemical composition of a medicinal plant is relevant to relate the phytochemical profile with the therapeutic properties. Thus, the present study is relevant and meets the scope of the journal.

Reading the manuscript showed many errors and adjustments that should be made in the text. The quality of writing is not good and, at times, it becomes a text that is heavy and confusing to read. In this way, I do not consider it to have the necessary quality for the publication of Plants. A comprehensive and rigorous review of the manuscript is required.

Response: We would like to thank you for your kind review and for taking the time and effort necessary to improve our manuscript. We found your comments very helpful and revised the manuscript accordingly.

Below are the tweaks and corrections.

- In line 21, replace “chlorogenic” with ‘chlorogenic acid’;

- On line 23, replace “3.3 to 6.5.” by “3.3 to 6.5%.”;

- On line 25, I suggest replacing "create" with "develop”;

- In lines 31-32, the term "European, American, American herbal" is confused. Do the authors intend to write “American and European Herbal Pharmacopoeias”? Please review.

Response: the indicated mistakes were corrected, the missing words were included, the names were checked, and the sentences were rearranged.

 - On line 42, the authors wrote "which is caused primarily by sesquiterpene lactones [13]". The study described in reference 13 contains pharmacological experiments on the plant extract only. There is a computer simulation of a constituent of the plant, which is a binding energy calculation. It does not allow concluding that it has pharmacological action. Therefore, the authors cannot claim that sesquiterpene lactones have antimigraine activity, based on this study. Please correct your writing.

Response: The reference was changed to: Lopresti, A.L.; Smith, S.S.; Drummond, P.D. Herbal treatments for migraine: A systematic review of randomised-controlled studies. Phytother. Res. 2020, 34, 2493–2517.

 - On lines 46-47 authors wrote "Over the past 10 years, biologically active additives based on this raw material have been developed in Western countries [9]". However, reference 9 is a literature review published in 2004. So it is wrong to write that it is about the last 10 years. Please update the reference or revise the sentence.

Response: the reference was updated: Goschorska, M.; Gutowska, I.; Baranowska-Bosiacka, I.; Barczak, K.; Chlubek, D. The Use of Antioxidants in the Treatment of Migraine. Antioxidants 2020, 9, 116.

 - In line 56, the term chrysanthemum is not sesquiterpene. It is the name of a genus of plant. Please correct.

Response: the mistake was corrected.

 - On line 57, replace " (chlorogenic, dicaffeoylquinic, chicoric" with " (chlorogenic acid, dicaffeoylquinic acid, chicoric acid".

Response: the replacement was made.

 - On line 59, correct "6-trimethylfether" and proofread all compound names in this paragraph. For example, tanetin is not 6-trimethyl ether ether; camphor, camphene, p-cymene, bornyl acetate are not essential oils. They are constituents of essential oils. It is very important that authors be more careful in writing this information. There are many errors in the Introduction of the manuscript. Please review the paragraphs rigorously.

Response: the compound names were checked. The corrections were made and the sentences in the introduction and other parts of the manuscript were rearranged.

 - On line 68, correct “tops was 0.18%, 0.09 in leaves”.

Response: the corrections were made.

Sincerely,

Lina Raudone on behalf of all authors.

Reviewer 5 Report

Article entitled “Phenolic Composition ….. Acids Profile” describes phenolic profiles and antioxidant activity of cultivated feverfew plants. In this study, authors identified eleven phenolic compounds and also reported that Hydroxycinnamic acids are predominate. With this phenolic profile, authors concluded that feverfew is rich in phenolic substances which can be explored further to develop new herbal medicines. My comments are : –

1.      Is phenolic content sufficient to propose a plant to be explored for herbal medicines?

2.      Table 1, sample 866: The total content of flavonoids 192 ± 0.03 %. How?

3.      Table 1: Present data for flavonoids and hydroxycinnamic acids content in quantitative values (for example microgram per gram of sample as given in Table 2) rather than comparative %age.

4.      Authors mentioned that they identify different phenolics compound in results. But there is no methodology for identification? How they identify different phenolics compound?

5.      Provide a clear methodology for the estimation of is given for Table 2 in Material & Methods.

6.      Authors performed HPLC, which is used for Qualitative and Quantitative Analysis with reference compounds. What was the criterion for the selection of reference compounds? How they so selective about reference compound? They should perform LC/MS or GC/MS and identify compounds as much as possible, then they have to perform HPLC for quantitation and other analysis.

7.      Based on phenolic profile, how authors proposed for herbal medicines?

8.      Authors should corelate environmental data of samples with metabolites (phenolic compound).   

Author Response

Article entitled “Phenolic Composition ….. Acids Profile” describes phenolic profiles and antioxidant activity of cultivated feverfew plants. In this study, authors identified eleven phenolic compounds and also reported that Hydroxycinnamic acids are predominate. With this phenolic profile, authors concluded that feverfew is rich in phenolic substances which can be explored further to develop new herbal medicines

Response: We would like to thank you for your kind review and for taking the time and effort necessary to improve our manuscript.

  1. Is phenolic content sufficient to propose a plant to be explored for herbal medicines?

Response: Phenolic compounds are one of the groups of specialized metabolites that are characteristic of the T. parthenium plants. They have been mainly associated with various health benefits due to their antioxidant and anti-inflammatory properties. The latter is important for the management of neuroinflammation in the case of migraines. Moreover, the therapeutic effects of plant crude extracts are determined by the interaction between components or certain groups of bioactive compounds and also may show additive or synergistic action. Consequently, the quality depends on the full phytochemical composition thus, the determination of phytoprofiles of principle and complementary groups of compounds is essential for the comprehensive characterization of herbal drugs. The combined action of flavonoids and sesquiterpene lactones in feverfew makes it a unique and potentially effective herbal remedy, particularly in the management of migraines, headaches, and other inflammatory conditions. Phenolic compounds could be analysed as a complementary group compared to primary sesquiterpene lactones. Plants synthesize phenolic compounds in response to environmental factors, such as UV radiation, pathogen attacks, herbivore predation, and abiotic stress. The composition of phenolic compounds in a plant can, therefore, reflect its ecological niche and adaptation strategies. Therefore, geographical profiling could elucidate chemical polymorphism and identify promising genotypes for industrial plantations. In conclusion, while phenolic content is an essential aspect to consider when exploring a plant for herbal medicines, it is crucial to also analyze the overall phytochemical composition and consider interactions between different compounds to ensure the effectiveness, consistency, and quality of herbal remedies.

  1. Table 1, sample 866: The total content of flavonoids 192 ± 0.03 %. How?

Response: Thank you for noting. The mistake was corrected.

  1. Table 1: Present data for flavonoids and hydroxycinnamic acids content in quantitative values (for example microgram per gram of sample as given in Table 2) rather than comparative %age.

Response: The total amounts were determined using the NPhU 2.0 monograph where the amounts are expressed in percentages. It's important to note that the total content assays provide an overall estimation of the flavonoid or hydroxycinnamic acid concentration in the herbal sample for the general comparisons between samples and the results of other studies. For a more comprehensive analysis, individual flavonoid compounds were quantified using high-performance liquid chromatography.

  1. Authors mentioned that they identify different phenolics compound in results. But there is no methodology for identification? How they identify different phenolics compound?

Response: The analyte and reference compound retention time, UV absorption spectra and spiking of samples were used for peak identification.

  1. Provide a clear methodology for the estimation of is given for Table 2 in Material & Methods.

Response: The calibration curves of reference compounds were constructed. Contents of phenolic acids were estimated at a wavelength of 325 nm, while the contents of flavonoids were estimated at a wavelength of 350 nm.

  1. Authors performed HPLC, which is used for Qualitative and Quantitative Analysis with reference compounds. What was the criterion for the selection of reference compounds? How they so selective about reference compound? They should perform LC/MS or GC/MS and identify compounds as much as possible, then they have to perform HPLC for quantitation and other analysis.

Response: Regarding the suggestion of using LC/MS or GC/MS to identify compounds before HPLC quantitation, this approach is indeed valuable and commonly employed in the comprehensive profiling and identification of complex mixtures. However, not all compounds identified with the MS detection system are available. We referred to previous studies and scientific literature to identify commonly used reference compounds for T. parthenium extracts. Our assay targeted the quantitative phenolic markers.

  1. Based on phenolic profile, how authors proposed for herbal medicines?

Response: The aim of the study was to investigate the phenolic compounds and antioxidant activity present in feverfew herbs collected from various regions of Ukraine. The oxidative stress and inflammation may play a role in triggering and exacerbating migraines in some individuals. The migraine management is multifaceted and different groups of compounds contribute to the activity of T. parthenium and substantiate its ethnopharmacological relevance. Furthermore, the coupled knowledge on principal groups of compounds, such as phenolic and terpenic ones could provide potential markers for standardization for the future development of medicines based on feverfew as both groups of compounds could contribute to the targeted pharmacological effect.

  1. Authors should corelate environmental data of samples with metabolites (phenolic compound).

Response: We sincerely appreciate your valuable feedback and thoughtful suggestion to correlate the environmental data of the samples with the phenolic compounds. We have provided the cluster analysis and the possible effects of specific parameters of climatic conditions characteristic to the defined areas. The comprehensive multiannual evaluation of environmental data to the profiles of specialized metabolites would be implemented in our further research.

Sincerely,

Lina Raudone on behalf of all authors.

Round 2

Reviewer 1 Report

not much improved.

Minor editing of English language required

Author Response

We would like to thank you for your kind review and for taking the time and effort necessary to improve our manuscript. The English language was revised. All changes have been indicated in red in the manuscript.

Sincerely,

Lina Raudone on behalf of all authors.

Reviewer 4 Report

Dear editor,

whereas corrections have been made to the text,

the manuscript is eligible for publication in Plants.

Author Response

Thank you very much. 

Reviewer 5 Report

Reply is satisfactory

Author Response

Thank you very much. 

Sincerely,

Lina Raudone on behalf of all authors.